# Three-Dimensional-Bioprinted Bioactive Glass/Cellulose Composite Scaffolds with Porous Structure towards Bone Tissue Engineering

**DOI:** 10.3390/polym15092226

**Published:** 2023-05-08

**Authors:** Lei Li, Pengfei Lu, Yuting Liu, Junhe Yang, Shengjuan Li

**Affiliations:** School of Materials and Chemistry, University of Shanghai for Science and Technology, 516 Jungong Road, Shanghai 200093, China

**Keywords:** 3D printing, biomaterials, lignocellulose, bioactive glass, biocompatibility, bone tissue engineering scaffold

## Abstract

In this study, three-dimensional (3D) bioactive glass/lignocellulose (BG/cellulose) composite scaffolds were successfully fabricated by the 3D-bioprinting technique with *N*-methylmorpholine-*N*-oxide (NMMO) as the ink solvent. The physical structure, morphology, mechanical properties, hydroxyapatite growth and cell response to the prepared BG/cellulose scaffolds were investigated. Scanning electron microscopy (SEM) images showed that the BG/cellulose scaffolds had uniform macropores of less than 400 μm with very rough surfaces. Such BG/cellulose scaffolds have excellent mechanical performance to resist compressive force in comparison with pure cellulose scaffolds and satisfy the strength requirement of human trabecular bone (2–12 MPa). Furthermore, BG significantly increased the excellent hydroxyapatite-forming capability of the cellulose scaffolds as indicated by the mineralization of the scaffolds in simulated body fluid (SBF). The BG/cellulose scaffolds showed low cytotoxicity to human bone marrow mesenchymal stem cells (hBMSCs) in the CCK8 assay. The cell viability reached maximum (percent of the control group) when the weight ratio of cellulose to BG was 2 in the scaffold. Therefore, the 3D-printed BG/cellulose scaffolds show a potential application in the field of bone tissue engineering.

## 1. Introduction

Natural bones comprise a ceramic phase mixed with protein and polysaccharides. Such a highly specialized organic–inorganic architecture provides a rigid state to support the torso, protect soft organs, store minerals and enable body motion [1]. Bone tissue shows unique hierarchical structure, with various constituents classified to each structure level, including macrostructure, microstructure, sub-microstructure, nanostructure and sub-nanostructure [2]. As the main part of bone, cortical bone encloses porous trabecular bone (i.e., cancellous or spongy bone) to optimize articular load transfer. Such composite materials and the porous structure of bones can provide load-bearing capability and flexibility of bone to protect it from fractures [3]. The compressive strength of human trabecular bone is in the range of 2–12 MPa [2]. Furthermore, through the interconnected porous network, bone can provide a nano- and micro-environment that facilitates cell growth [2]. Therefore, the porous structure is critical for bone to realize both mechanical and biological capabilities.

Bone itself has an excellent regenerative capacity to heal from small defects. However, many bone diseases can result in serious bone tissue deficits that cannot self-heal, even when using traditional intervention [4]. Treatments of bone deficits (missing bone or bone tumors) are a costly problem in healthcare. Resection of tumors and functional reconstruction of bones followed by postoperative radiation or chemotherapy is a common technique [5]. Bones are the second most transplanted tissue after blood per year. Materials to replace missing bones from the patient (autologous bone grafting) are limited by quantities, shapes and sizes, although they are optimized. Alternatives from another human (allogenous grafting) or from another species (xenografting) can cause graft rejection and pathogen transmission [6]. A bone tissue engineering (TE) scaffold is an efficient approach to extend TE applications in past decades [7]. The three-dimensional biomaterial scaffold can provide structural support for cell attachment, spreading, migration, proliferation and differentiation [8]. It is necessary for bone TE to possess a network of interconnected pores to mimic bone functions of cell migration and nutrient transport for the cells. Pore sizes larger than 300 μm are efficient to support bone growth [9].

Synthetic or natural polymers are extensively employed as biomaterials in the fabrication of extracellular matrix to mimic advanced bone TE scaffolds, including polyhydroxyalkanoates (PHAs), polyurethanes (PURs), polyphosphazenes, collagen, fibrin and hyaluronic acid (HA). Cellulose is the most abundant and renewable biopolymer on earth with fascinating structure and properties, which have attracted tremendous attention to exploit its novel functions and extend its applications. The primary source of cellulose is the lignocellulosic material from trees. Other plants also can contribute cellulose, as can algae, bacterial biosynthesis and chemosynthesis [6]. Cellulose has repeated 𝛽-(1,4) linked D-glucose building blocks on its linear chains, which provide cellulose with high hydrophilicity, natural biodegradability, facile chemical modification and a wide range of fiber morphologies [10].

Inorganic materials are also regarded as potential bioactive materials for bone TE, such as calcium phosphates, hydroxyapatite and bioactive glass (BG). BG is a silicate system comprising SiO_2_, Na_2_O, CaO and P_2_O_5_. Its surface layer of calcium-deficient carbonated phosphate can integrate bones via chemical bonds and form a carbonated Hap layer on the glass surface when it is implanted into the animal body or set in biological fluids [11,12]. Having excellent bioactivity, biocompatibility, osteogenic and angiogenic effects, BG can therefore function as bone tissue engineering regeneration material, grow biofactors (cells, genes and/or proteins) in physiological fluids, repair bone and tooth defects, and even has the capability to heal soft tissue or turn to new blood vessels [13]. In particular, BG 45S5 comprising 45% SiO_2_, 24.5% Na_2_O, 24.5%CaO and 6% P_2_O_5_ by weight is often employed as bone filler and in clinical treatment of bone diseases [14]. However, pure BG is brittle, very hard and non-flexible. BG employed as TE scaffolding is often in the form of coating, since porous structures normally show very poor mechanical properties with respect to healthy bone, although it is required in bone TE scaffolds. It is still attracting great interest to fabricate BG porous scaffolds. Polymer–inorganic composite materials have already been developed to solve both the mechanical and bioactive problems of TE scaffolds [6,15]. Polyvinyl alcohol (PVA) and microfibrillated cellulose (MFC) were reported to reinforce the mechanical properties of BG-based scaffolds [16]. The BG degradation properties can also be adjusted according to a selected application [17]. The composite scaffolds prepared from natural polymers and BG are proposed to balance the compressive strength and toughness.

Three-dimensional (3D) printing is a developing technology which has been used in a wide range of fields for both personal and industrial manufacturing. Being capable of rapidly prototyping a wide range of object geometries, 3D printing is making huge impacts in many fields, such as aerospace, boats, automobiles, construction, food production, biomedical and tissue engineering. This is because 3D printing can precisely and integrally manufact highly complicated structures through computer and digital techniques without complex tools or additional connecting fittings [18]. In the next decade, $2 billion will be invested in the biomedical field to support 3D-printing techniques [19]. A 3D-bioprinting technique is predominantly used to produce customizable TE scaffolds with an interconnected porous structure, which shows the capability of transplanting a biofactor within the scaffold [20]. Many types of materials, including polymers, metal powders, ceramic, sand, wax and certain gel-like biomaterials have been successfully 3D-printed. It is worth noting that currently popular 3D-printing polymeric materials like polylactic acid (PLA), acrylonitrile butadiene styrene (ABS) and nylon may cause potential hazards. Volatile organic compounds (VOCs) and aerosols can be generated by these materials in the 3D-printing process that are harmful to humans. It is important to explore low-emitting and less toxic 3D-printing materials. Unlike synthetic polymers, natural polymers are renewable and biodegradable. It is attractive to develop 3D-printable biopolymers with high-quality performance. Many natural polymer hydrogels have demonstrated their potential as 3D-printing materials. Cellulosic derivative materials have also been investigated to fabricate 3D objects. Specifically, cellulose is a unique biomaterial, which can provide an opportunity to prepare 3D-printing materials cheaply and safely due to its low cost and abundant resources. However, it is still a huge challenge to practically realize lignocellulose 3D printing because no common solvent can dissolve cellulose at room temperature [21].

In our prior study, pure cellulose scaffolds with tough controlled interconnected porous structures were successfully 3D-printed using ink with dissolved cellulose in *N*-methylmorpholine-*N*-oxide (NMMO) solution [22]. The present work shows the further developed novel BG/cellulose TE scaffolds prepared by the same 3D-bioprinting technique, where cellulose acts as the main material to be 3D-printed into the designed shape, while BG acts as a stiffness promoter with hydroxyapatite growth facilitation. The compressive strength of the cellulose scaffold is enhanced significantly when BG is supplemented into the composite, which satisfies the mechanical range of a natural bone. Furthermore, the hydroxyapatite deposition capability of the composite scaffolds is dependent on the weight ratio of cellulose to BG. The low cytotoxicity of the BG/cellulose scaffold is also demonstrated in the present work.

## 2. Materials and Methods

### 2.1. Materials

Dissolving pulps (cellulose content >90%) were obtained from Shandong Yamei Sci-Tech Co. Ltd (Binzhou, China) with a dynamic viscosity of 19 mPa∙s. NMMO (50 wt.% H_2_O) and dimethylsulfoxide (DMSO) were purchased from HWRK Chemical Reagent Co. LTD (Beijing, China). BG 45S5 powders were obtained from Kunshan Overseas Chinese Technology New Materials Co. Ltd. (Kunshan, China). The mean diameter of the BG particles was approximately 500 nanometers, with a size scope of 0.1–10 μm (Figure 1d).

### 2.2. Dissolution of Dissolved Pulp

The pulp was dissolved in NMMO (50 wt.% H_2_O) according to our reported protocol [22]. NMMO and DMSO are efficient solvents frequently used in the pulp-dissolution process to dissolve cellulose. After adding 5 g dissolving pulps in 100 mL NMMO, the temperature of the cellulose/NMMO solution was raised to 115 °C under vigorous stirring. When a yellow transparent solution was obtained, 10 mL DMSO was added dropwise to adjust viscosity of the solution before the solution temperature was reduced to 70 °C. Meanwhile, BG powders were sieved through 400 mesh sieves and added into the solution. The cellulose/NMMO solution was vigorously stirred until BG powder was homogeneously distributed in the solution. Samples obtained from various amounts of BG powders in the next step were denoted as 5CELL-BG, 2CELL-BG and 1CELL-BG for the ratio of cellulose/BG = 5:1, 2:1 and 1:1 in weight, respectively.

### 2.3. Process of 3D Printing BG/Cellulose Scaffolds

The BG/cellulose/NMMO solution at 70 °C was loaded into the stainless ink extrusion cartridge on the 4th 3D bioplotter^TM^ printing device (Envision TEC GmbH, Gladbeck, Germany). Then, cylindrical block models (ϕ = 10 mm) were loaded on the Bioplotter CAD/CAM software. Bioprinting was performed by keeping the temperature of the cartridge at 70 °C, then the ink of BG/cellulose/NMMO within the cartridge was extruded to form a gel fiber. Thereafter, the gel fibers could freely pile up layer-by-layer to at least 1 cm in height at room temperature. The printing direction was changed by plotting fibers with 0 and 90 degrees in turn between two successive layers. The dosing pressure to the syringe pump was 3.0 bar, and the speed of the dispensing unit was 3–5 mm/s. The nozzle size was 0.25 mm. When the scaffold reached a height of 10 mm, the 3D-printing process was finished. The 3D-printed BG/cellulose/NMMO scaffold was immediately placed into Milli-Q water to remove NMMO and DMSO from the scaffold [23]. Subsequently, the prepared BG/cellulose hydrogel scaffold was frozen at −70 °C for 12 hours and then freeze-dried into a tough BG/cellulose scaffold.

### 2.4. Characterization

#### 2.4.1. XRD and SEM

The 3D-printed BG/cellulose scaffold was characterized by X-ray diffraction (XRD) (Bruker D8 Advance, Bruce Inc., Bremen, Germany) with Cu Kα as the radiation source (λ = 1.5406 Å). The operation voltage was 40 kV with a current of 40 mA for the X-ray beam. Scanning was in a range of 2θ = 20–80° and at a speed of 5.0°/min. Field scanning electron microscopy (SEM) (Quanta TM 450 FEG, FEI, Hillsboro, OR, USA) was employed to investigate the morphologies of the BG/cellulose scaffolds at an electron acceleration voltage of 30 kV. The Ca/P ratio was determined by energy-dispersive X-ray spectroscopy (EDS).

#### 2.4.2. Porosity

The BG/cellulose scaffold does not swell obviously. As such, the porosities of the 3D-printed BG/cellulose scaffolds were determined using Archimedes’ principle. Porosity is calculated by weighing the scaffold in the air and in the water, respectively. The formulation is: P = (W_sat_ − W_dry_)/(W_sat_ − W_sus_) × 100%, where W_dry_ is the dry weight of the composite scaffolds in air, W_sus_ is the weight of the scaffolds suspended in water, and W_sat_ is the weight of scaffolds saturated with water.

#### 2.4.3. Contact Angle Measurement

The equilibrium contact angles of water droplets on the pure cellulose and BG/cellulose material surfaces regenerated from NMMO were recorded when the contact angle reached a constant (θ_E_), respectively. The solutions of cellulose/NMMO and BG/cellulose/NMMO (cellulose:BG = 2:1) were casted on a silicon substrate, respectively. Thereafter, the films were regenerated by immersing them in water for 10 seconds to remove NMMO and then heated with a hair dryer for 5 seconds to stabilize the films on the silicon substrate. Contact angle measurements were performed at room temperature on a custom-built contact angle system. The liquid probe was a 5 µL Milli-Q water drop of 18.2 MΩ·cm. The water contact angle was measured five times at different positions on the sample surface. Average values and statistical errors were calculated.

#### 2.4.4. Mechanical Testing

The compressive strengths of the 3D-printed cylindric scaffolds (ϕ = 1 cm) were tested on a Zwick static-materials testing machine (5 kN) at a cross head speed of 0.5 mm/min. The load was applied to the top plane of the scaffolds when the compressive test began. The compressive strength was determined according to the maximum load of the stress-strain curve before the collapse of the scaffolds. Each type of scaffold was tested five times (n = 5) using the selected typical samples. Then the average values and standard deviations were calculated for the four types of scaffolds.

#### 2.4.5. In Vitro Bioactivity Assessment

The in vitro bioactivity test was conducted according to a standard procedure reported by Kokubo et al. [24]. The protocol of hydroxyapatite formation on scaffolds in simulated body fluid (SBF) is described as follows: SBF is a solution with ion contents similar to those in human blood plasma. The BG/cellulose scaffolds were immersed in 50 mL SBF and stored in a shaking incubator at 37 °C. The 3D printed BG/cellulose scaffolds (diameter of 10 mm and height of 10 mm) were immersed in SBF at a ratio of V_SBF_/W_scaffold_ = 200 ml/g and stored in a shaking incubator at 37 °C and 90 rpm for a period of several days. The SBF solution was not refilled in the culture period. In the present study, SEM and EDS images are provided to demonstrate the apatite deposition on the surfaces of all BG/cellulose scaffolds after immersion in SBF for 3 days.

#### 2.4.6. In Vivo Cytotoxicity Assessment

Before cell seeding, the scaffolds were sterilized under ultraviolet light, and then pre-wetted using culture medium in 24-well culture plate for more than 48 h.

The cell cytotoxicity of the BG/cellulose scaffolds was assessed via a Cell Counting Kit-8 (CCK8) assay and human bone marrow mesenchymal stem cells (hBMSCs). The protocol of the CCK8 assay is described in the literature [25]; briefly, 360 μL of culture medium and 40 μL of CCK-8 solution were added to each culture well on the 1st, 4th and 7th days and incubated at 37 °C. After 4 h, an aliquot of 100 μL was taken out and transferred to a new 96-well plate to measure its light absorbance at 450 nm through a microplate reader (Bio-Rad 680, Bio-Rad, Hercules, CA, USA). Readings under the Bio-Rad 680 were done in triplicate. The results of light absorbance were expressed as the optical density (OD) values minus the absorbance of the blank wells. The blank control was used with the same volume of culture medium without the addition of the prepared materials. The cells were incubated in the blank control, as well as the 24-well plate with the addition of the pure cellulose and the BG/cellulose scaffolds, respectively. In addition, the cell viability (% of the control) can be assessed by comparing the number of the live cells for each scaffold with the number of live cells for the control.

A statistical analysis was performed to express all data as means with standard deviations. LSD analysis was used to compare pairs after one-way analysis of variance (ANOVA). Statistical significance was defined as * *p* < 0.05, ** *p* < 0.01, and *** *p* < 0.001.

## 3. Results and Discussion

### 3.1. 3D-Printed BG/Cellulose Scaffolds before and after Freeze-Drying

The fresh 3D-printed BG/cellulose scaffold in a cylindrical shape with porous structure is presented in Figure 1a. Both the diameter and height of the scaffold are approximately 10 mm. It is worth noting that the scaffold does not obviously swell in water. Figure 1b shows the BG/cellulose scaffold after freeze-drying, which has slight deformation due to water removal, although it can maintain a tough shape and porous structure. An SEM image of the BG particles is presented in Figure 1c. BG particles were found to have a size range between 100 nm and 10 μm, with a main particle size of less than 1 μm according to the histogram of the size distribution of all BG particles in Figure 1d. Figure 1e shows the histogram of the BG particle sizes that statistically distribute between 100 nm and 1 μm. The average diameter (AD) of the BG particles is 489.1 ± 155.1 nm.

### 3.2. XRD Analyses

Figure 2 presents the XRD patterns of the pure BG powder, the pure cellulose and BG/cellulose scaffolds by 3D printing with different BG/cellulose ratios. The XRD pattern (a) shows one diffraction peak at 32° for the pure BG 45S5 powder, which cannot be observed in other patterns in Figure 2. The characteristic peaks of cellulose II phase are clearly observed in the XRD pattern (b), including 12°, 20.1° and 21.7° [8]. The broad diffraction peak centered at approximately 17° is observed in the 5CELL-1BG XRD pattern (c), which is assigned to amorphous cellulose phase. Therefore, the addition of BG affected the regeneration of crystalline cellulose phase from the BG/cellulose/NMMO intermediate product. Along with the enhancement of the BG amount in the composite scaffold, the crystallinity of cellulose increased significantly as the XRD pattern (d) (2CELL-1BG) and pattern (e) (1CELL-1BG) demonstrate. The reason is assumed to be the accumulation of BG due to the larger amount and its separation from the BG/cellulose complexes.

### 3.3. FTIR Analyses

The materials of cellulose, pure BG and BG/cellulose scaffold before and after immersion in SBF were investigated by using FTIR. Typical FTIR spectra of the above materials are compared in Figure 3. It is shown in Figure 3a that the spectrum of the BG/cellulose scaffold before SBF immersion is the superposition of the IR signals of cellulose and BG, including strong signals of the crosslinked ≡Si–O–Si≡ at approximately 1030 cm^−1^, and Si-NBO (non-bridge oxygen absorption) between 750 and 1000 cm^−1^ [26]. New bands in the spectrum of BG/cellulose after SBF immersion are clearly observed in the range of 1600–400 cm^−1^. Such bands are marked in Figure 3b by one band at approximately 873 cm^−1^ and the dual broad bands at 1420–1450 cm^−1^ to denote the stretching vibration of the C–O bond in formed carbonated hydroxyapatite (HCA). In addition, two bands at 566 cm^−1^ and 603 cm^−1^ arise due to the P–O bending vibrations in crystalline hydroxyapatite (HA) [27].

### 3.4. Mechanical Tests

Figure 4(a1) shows the compressive strength of pure cellulose, 5CELL-1BG, 2CELL-1BG and 1CELL-1BG scaffolds with the yield point values of 4.54 ± 1.14, 10.45 ± 5.80, 25.66 ± 2.78 and 27.0 ± 1.65 MPa, respectively. More addition of BG greatly increased the compressive strength and the mechanical stability of the 3D-printed cellulose/BG scaffolds. The increased compressive strength may be attributed to the formation of the stable 3D organic/inorganic network when the BG powders were mixed and distributed over the crystalline cellulose chains when the uniform and continuous pore structure was 3D-printed. Figure 4(a2) presents the compressive strength–strain curves of the four 3D-printed scaffolds. The maximum compressive strength increased and the compressive strain decreased when BG content increased in the composite scaffold. Figure 4(b1) presents the tensile stress of the pure cellulose, 5CELL-1BG, 2CELL-1BG and 1CELL-1BG scaffolds with values of 2.16 ± 0.64, 1.09 ± 0.13, 1.23 ± 0.31 and 1.15 ± 0.15 MPa, respectively. A significantly lower tensile stress of the cellulose scaffold was observed after the addition of the BG powder, which corresponds to more amorphous cellulose phase in the scaffold according to the XRD analysis. Nevertheless, the differences among the tensile stress values of the BG/cellulose scaffolds are small. It is the degree of the scaffold deformation (strain) that reduced greatly when more BG powder was added to the composite scaffolds (Figure 4(b2)).

### 3.5. Surface Hydrophilicity Measurement

Water contact angle was measured to evaluate the surface hydrophilicity of the pure cellulose and the BG/cellulose scaffolds. Photos of a water droplet on the pure cellulose and the BG/cellulose surfaces are presented in Figure 5a and Figure 5b, respectively. The pure cellulose showed a typical low contact angle of approximately 40°. The contact angle of BG/cellulose changed to approximately 34° due to the intrinsic hydrophilic property of BG 45S5. It is promising that the BG/cellulose surface will have better bioactivity and interaction with cells.

### 3.6. Analyses of SEM Images

Figure 6 shows SEM images of the 3D-printed pure cellulose (a1), 5CELL-1BG (b1), 2CELL-1BG (c1) and 1CELL-1BG (d1) scaffolds with the SEM image of the strand surface of each corresponding scaffold (a2, b2, c2 and d2), respectively. All scaffolds have ordered interconnected porous structures with approximately 200 μm fiber spacing without obvious strand damage. With the same 3D-printing parameters, the 3D-printed cellulose scaffold has obvious smaller pore size than the BG/cellulose composite scaffold. The pure cellulose can undergo shape transformation in the freeze-drying process. The cellulose scaffold with BG was relatively stable in the freeze-drying process. It is observed that the fibers fused in the cross points due to the self-weight when the ink was extruded from the cartridge to form the intermediates. As shown in Figure 6(a2,b2,c2,d2), the strand surface roughness increased accordingly with more BG contents in the scaffold. The SEM images in Figure 6(c2,d2) show that the fuzzy morphologies with hierarchical structures of the BG/cellulose fibers were realized when BG powder was added to cellulose. Such irregular morphologies are clearly observed (Figure 6(c2,d2)), in contrast to the relatively smooth surface in Figure 6(a2).

Figure 6(a3,b3,c3,d3) presents SEM images of apatite deposition on the scaffold surfaces after 3-day simulated body fluid (SBF) immersion. All scaffolds maintained their mechanical stability well without damage to the strands and are therefore suitable for bone repair. Nevertheless, obvious differences are observed from the scattered hydroxyapatite particles over the pure cellulose (Figure 6(a3)) and 5CELL-1BG surfaces (Figure 6(b3)) to the densely covered hydroxyapatite layers on 2CELL-1BG (Figure 6(c3)) and 1CELL-1BG (Figure 6(d3)). The Ca/P molar ratios were calculated according to the EDS spectra (Figure 6(a4, b4, c4, d4)) to be 0.67 (pure cellulose), 0.95 (5CELL-1BG), 1.34 (2CELL-1BG) and 2.47 (1CELL-1BG), respectively. The 2CELL-1BG scaffolds showed the best Ca/P ratio value among the four prepared types of BG/cellulose scaffolds in comparison to 1.67 of hydroxyapatite and indicated their sufficient calcium contents in the large particle layers. As such, the weight ratio between cellulose and BG is found to be an important parameter to construct the optimum BG/cellulose composite scaffolds, which also may affect the bioactivity and biocompatibility of the BG/cellulose scaffolds.

### 3.7. Porosity Calculation

Figure 7 shows that the porosities of the BG/cellulose scaffold were measured to be 82.3 ± 1.6% (5CELL-1BG), 84.3 ± 2.3% (2CELL-1BG) and 85.2 ± 1.2% (1CELL-1BG), higher than 66.0 ± 10.6% of the pure cellulose scaffold. The porosity of the pure cellulose scaffold is clearly different from the BG/cellulose scaffolds. This is because the pure cellulose can obviously swell in water; however, the BG/cellulose scaffolds did not swell significantly in water. A more precise method to measure the porosity of the cellulose scaffold may be needed.

### 3.8. CCK8 Assay

Cell viability was determined by a CCK8 assay with cell proliferation of hBMSCs on the four types of scaffolds with the same scaffold parameters, including pure cellulose regenerated from NMMO, 5CELL-1BG, 2CELL-1BG and 1BG-CELL, and one blank control group. The blank control was set by the same environment and cells as other samples except that no scaffold was added. The results of cell viability (% of the control) after a 7-day cell culture are presented in Figure 8. The pure cellulose, 5CELL-1BG and 1CELL-1BG scaffolds behaved similarly to the control group only with a small enhancement of cell viability, which demonstrates that the prepared BG/cellulose scaffolds can stimulate cell proliferation and suggests their low cytotoxicity with/without the addition of BG to cellulose. This is because of excellent biocompatibilities of both cellulose and BG [10]. Specifically, the 2CELL-1BG scaffolds stimulated cell viability much greater up to approximately 314% ± 42% with a significant difference (*p* < 0.05, marked with * in Figure 8). Such performance is consistent with the Ca/P ratio of the four types of scaffolds in the hydroxyapatite formation experiment in SBF. Further investigation regarding this phenomenon will be conducted in the next study stage. 

## 4. Conclusions

In this study, the BG/cellulose scaffolds were fabricated through a 3D bioprinting technique without the necessity of any binding agent. The prepared BG/cellulose scaffolds have controlled spatial interconnected porous structure and a very rough surface. The addition of BG decreased the crystallinity of the cellulose scaffold while enhancing its compressive strength to meet the requirement of human trabecular bone (2–12 MPa). BG facilitated the hydroxyapatite-formation on the BG/cellulose scaffold when immersed in SBF. The 2CELL-1BG scaffold showed the optimum Ca/P ratio close to 1.67 of hydroxyapatite formation. The results of the CCK8 assay demonstrated the low cytotoxicity of all 3D-bioprinted BG/cellulose scaffolds. The best cell viability (% of the control) appeared for the BG/cellulose scaffold with the ratio of cellulose to BG of 2 (2CELL-1BG), which is consistent with the Ca/P ratio preformation in the hydroxyapatite formation experiment. In summary, the BG/cellulose scaffolds could be economically 3D bioprinted into alternatives in the field of bone tissue engineering.

## Figures and Tables

**Figure 1 polymers-15-02226-f001:**
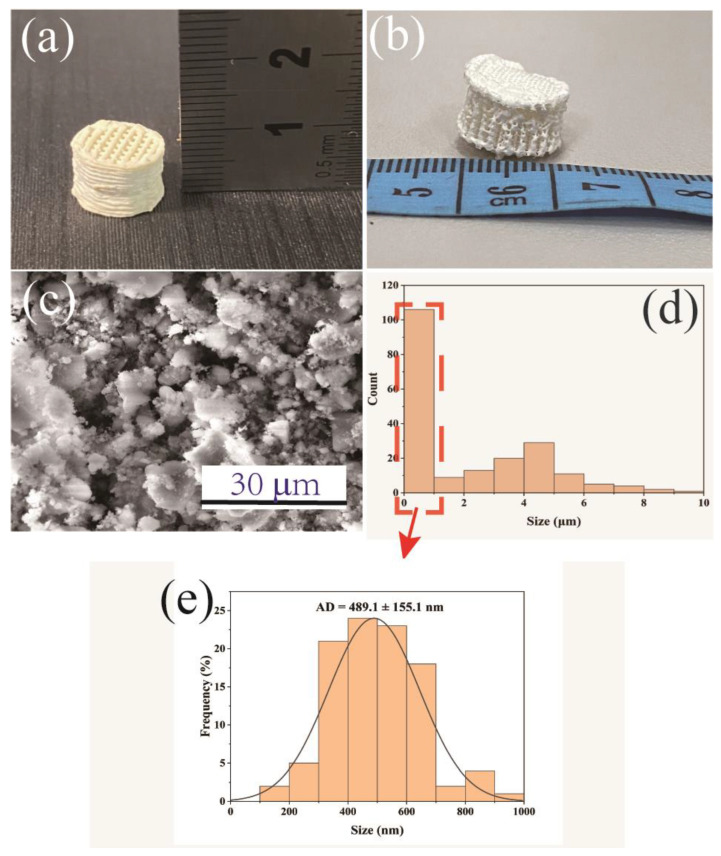
A typical 3D-printed BG/cellulose scaffold with approximately 1 cm height (**a**) before freeze-drying and (**b**) after freeze-drying; (**c**) SEM image of bioactive glass 45S5; (**d**) histogram of the size distribution for all BG particles; and (**e**) histogram of the BG particle sizes that statistically distribute between 100 nm and 1 μm.

**Figure 2 polymers-15-02226-f002:**
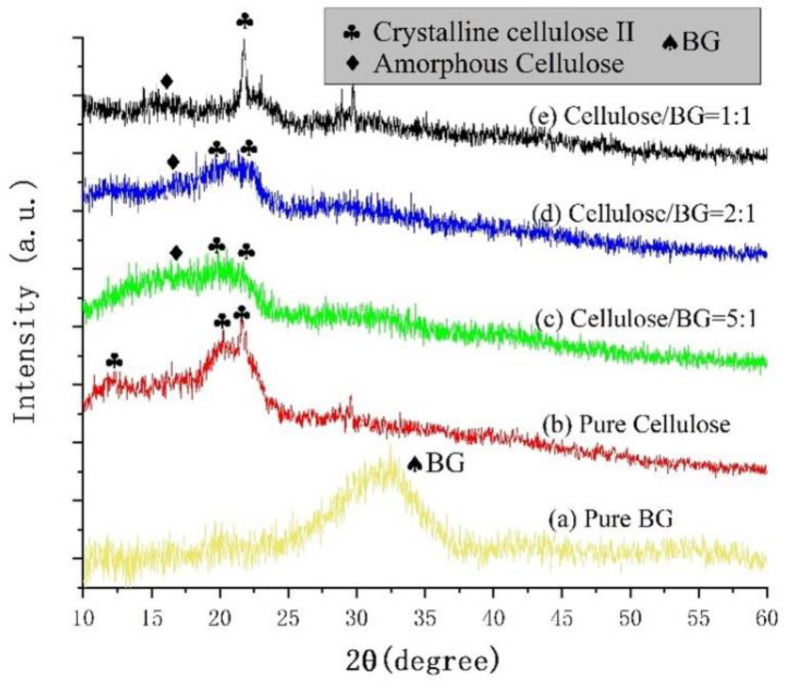
XRD patterns of the 3D-printed scaffolds: (**a**) pure BG; (**b**) pure cellulose; (**c**) cellulose:BG = 5:1 (5CELL-1BG); (**d**) cellulose:BG = 2:1 (2CELL-1BG) and (**e**) cellulose:BG = 1:1 (1CELL-1BG).

**Figure 3 polymers-15-02226-f003:**
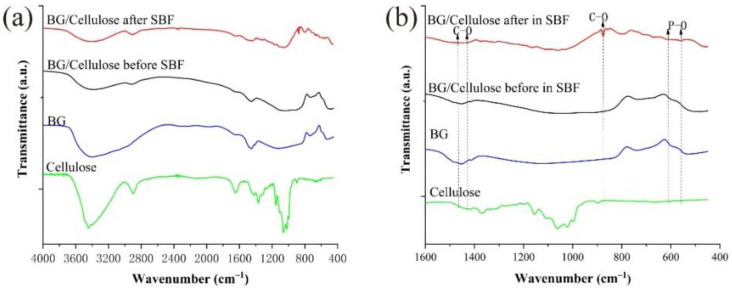
FTIR spectra of cellulose, BG and the 3D-printed BG/cellulose scaffolds in range of (**a**) 4000–400 cm^−1^ and (**b**) 1600–400 cm^−1^.

**Figure 4 polymers-15-02226-f004:**
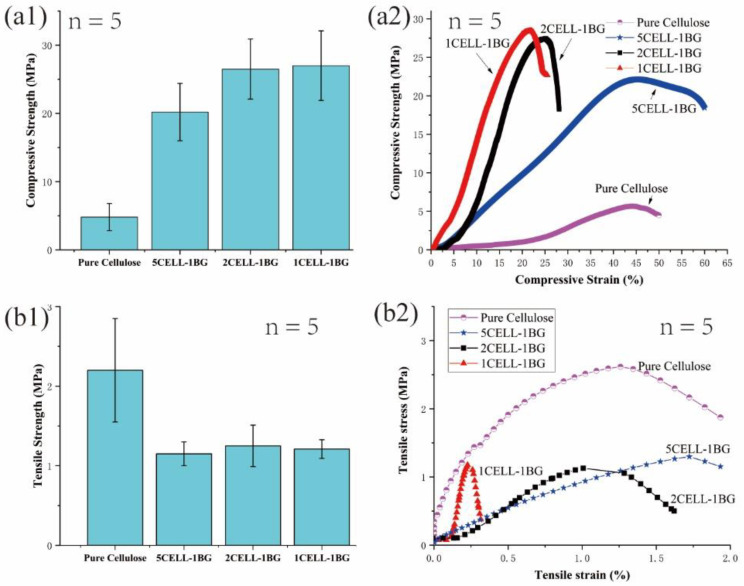
(**a1**) Compressive strength histogram and (**a2**) the corresponding strength–strain curves; (**b1**) Tensile strength histogram and (**b2**) the corresponding stress–strain curves of pure cellulose, 5CELL-1BG, 2CELL-1BG and 1CELL-1BG; n = 5 denotes five samples were measured for each material.

**Figure 5 polymers-15-02226-f005:**
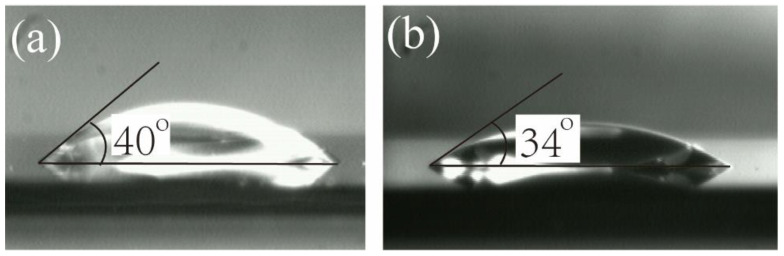
Photos of water droplet on (**a**) the cellulose surface and (**b**) the BG/cellulose surface. The contact angle of (**a**) is 40° while that of (**b**) is 34°.

**Figure 6 polymers-15-02226-f006:**
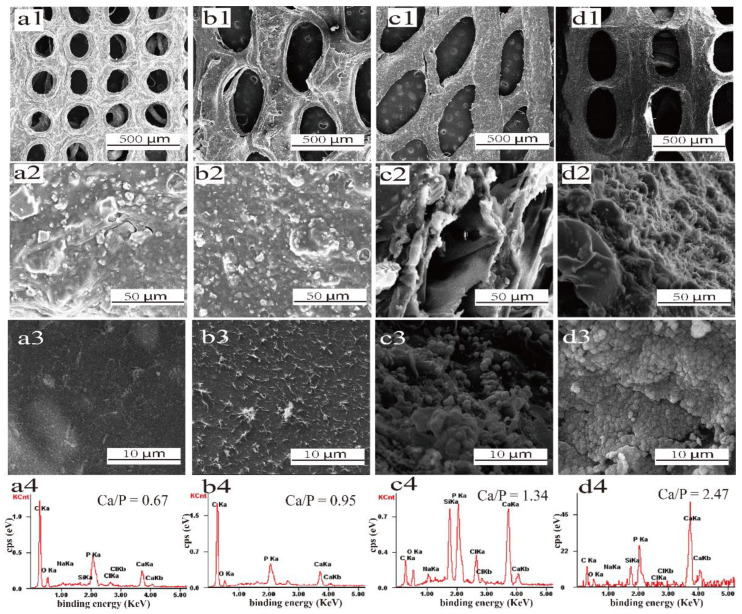
SEM images of one 3D-printed pure cellulose scaffold (**a1**–**a3**), three scaffolds of cellulose:BG = 5:1 (**b1**,**b2**), cellulose:BG = 2:1 (**c1**,**c2**) and cellulose:BG = 1:1 (**d1**,**d2**) under different magnifications; SEM images under different magnifications and EDS of the scaffolds after immersing in SBF for three days: pure cellulose (**a3**,**a4**), cellulose:BG = 5:1 (**b3**,**b4**), cellulose:BG = 2:1 (**c3**,**c4**) and cellulose:BG = 1:1 (**d3**,**d4**).

**Figure 7 polymers-15-02226-f007:**
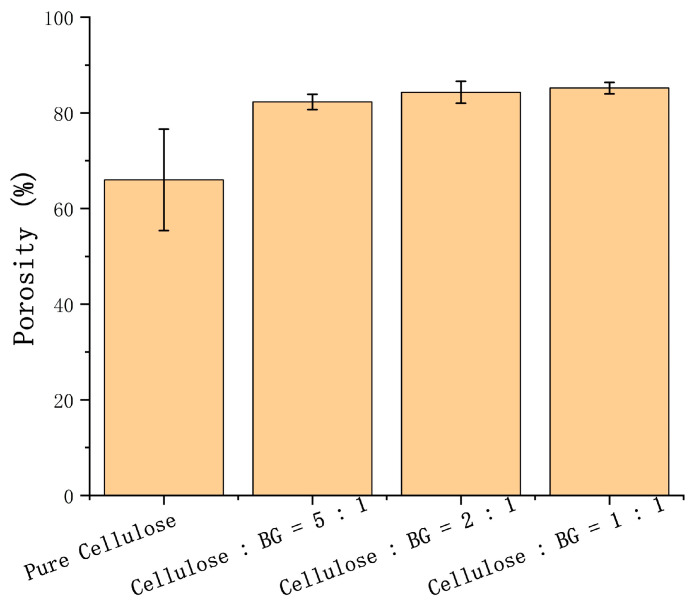
Histogram of porosity for pure cellulose, cellulose:BG = 5:1, cellulose:BG = 2:1 and cellulose:BG = 1:1 (*w*/*w*).

**Figure 8 polymers-15-02226-f008:**
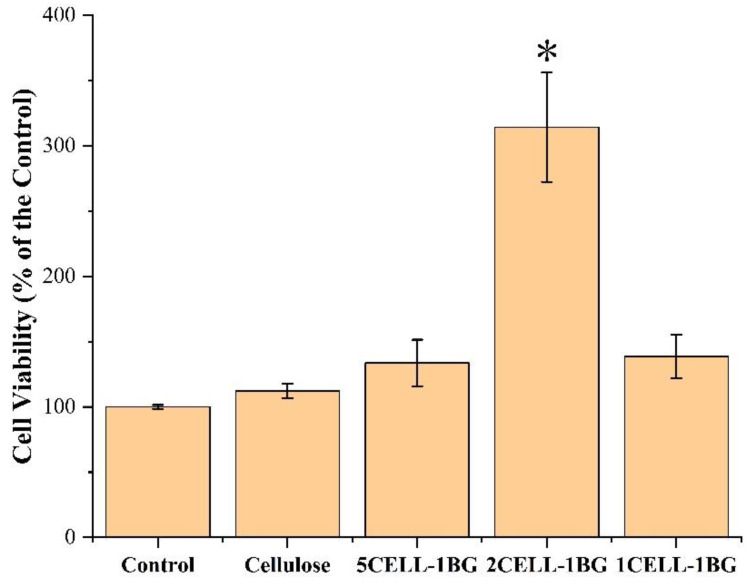
CCK8 assay of the 3D-printed scaffolds of pure cellulose and BG/cellulose composites with ratios of 5/1(5CELL-1BG), 2/1 (2CELL-1BG) and 1/1 (1CELL-1BG). * means statistical significance defined as *p* < 0.05.

## Data Availability

Not applicable.

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
