# Peer review of "Three-Dimensional-Bioprinted Bioactive Glass/Cellulose Composite Scaffolds with Porous Structure towards Bone Tissue Engineering"

_polymers, 2023, doi:10.3390/polym15092226_

Round 1

Reviewer 1 Report

In this study, authors fabricated the three-dimensional (3D) bioactive glass/lignocellulose (BG/cellulose) composite scaffolds by the 3D-bioprinting technique with N-Methylmorpholine-N-Oxide (NMMO) as the ink solvent. The physical structure, morphology, mechanical properties, hydroxyapatite growth and cell response to the as-prepared BG/cellulose scaffolds were investigated. The scanning electron microscopy (SEM) images showed that the BG/cellulose scaffolds had uniform macropores of less than 400 μm with highly rough surfaces. Meanwhile, such BG/cellulose scaffolds have excellent mechanical performance to resist compressive force in comparison with the pure cellulose scaffolds and satisfy the strength requirement of human trabecular bone (2-12 MPa). Overall, I think it is the extended research paper of Reference 8. Also, there are many issues in the main article, thus, I think it should undergo at least one round of major revisions.

1. In Supporting Information. Figure S1-d caption+ description. Note that the histogram of BG particle size distribution between 100 nm and 1 μm. Not 10 μm! Please carefully check it.

2. Page 6, line 189. Caption of Figure 4. Please delete “3.2. Mechanical Tests”

3. Many writing issues. Page 2, H2O should be H2O (appear several times in the main article). Page 3. “…. 12o,20.1o…..” Why did the o have the underline? Please carefully check it.

4. Note that “When the 3D printing process was finished, the 3D printed BG/cellulose/NMMO scaffold was immediately put into Milli-Q water to remove NMMO and DMSO from the scaffold”, did authors consider the DMSO residues? Also, when the solvent was exchanged, the porous structures actually may also change. Experimentally, to fully remove the organic solvent, three-time DI water washing is required. Thus, I recommended authors think about the experiment details, since it will later affect the biocompatibility.

5. I think some FTIR spectra should be listed.

6. Figure 2. There are two b.1. Please replace one with b.2.

7. Figure 4. P-value should be added, since these data must be conducted on the same day.

Please check my raised comments. Some writing should be revised.

Author Response

#1

In this study, authors fabricated the three-dimensional (3D) bioactive glass/lignocellulose (BG/cellulose) composite scaffolds by the 3D-bioprinting technique with N-Methylmorpholine-N-Oxide (NMMO) as the ink solvent. The physical structure, morphology, mechanical properties, hydroxyapatite growth and cell response to the as-prepared BG/cellulose scaffolds were investigated. The scanning electron microscopy (SEM) images showed that the BG/cellulose scaffolds had uniform macropores of less than 400 μm with highly rough surfaces. Meanwhile, such BG/cellulose scaffolds have excellent mechanical performance to resist compressive force in comparison with the pure cellulose scaffolds and satisfy the strength requirement of human trabecular bone (2-12 MPa). Overall, I think it is the extended research paper of Reference 8. Also, there are many issues in the main article, thus, I think it should undergo at least one round of major revisions.

  1. In Supporting Information. Figure S1-d caption+ description. Note that the histogram of BG particle size distribution between 100 nm and 1 μm. Not 10 μm! Please carefully check it.

Authors Reply: Thank the reviewer for the comment. The range of the BG particle size distribution has been changed to “between 100nm and 1 μm”. Meanwhile, all materials in Supporting Information have been transferred to the main texts of the manuscript. The histogram of BG particle size distribution is in Figure 1d in the revised manuscript.

  1. Page 6, line 189. Caption of Figure 4. Please delete “3.2. Mechanical Tests”

Authors Reply: Thank the reviewer for the comment. Authors believe the revised manuscript has been submitted after the requirement of the editor for more data and content. The error of “3.2 Mechanical Tests” has been deleted in the revised manuscript.

  1. Many writing issues. Page 2, H2O should be H2O (appear several times in the main article). Page 3. “…. 12o,20.1o…..” Why did the ohave the underline? Please carefully check it.

Authors Reply: Thank the reviewer for the comments. All the errors are caused by the font in the template. All mentioned errors have been corrected.

  1. Note that “When the 3D printing process was finished, the 3D printed BG/cellulose/NMMO scaffold was immediately put into Milli-Q water to remove NMMO and DMSO from the scaffold”, did authors consider the DMSO residues? Also, when the solvent was exchanged, the porous structures actually may also change. Experimentally, to fully remove the organic solvent, three-time DI water washing is required. Thus, I recommended authors think about the experiment details, since it will later affect the biocompatibility.

Authors Reply: The reviewer is right. Actually, NMMO and DMSO was extracted from the scaffold via the Milli-Q water at least ten times with replaced clean Milli-Q water after each extraction. The aim is to completely remove NMMO and DMSO. The porous structure of the 3D printed BG/cellulose scaffold did not change obviously during the NMMO/DMSO extraction. Nevertheless, the porous structure of the scaffold may transform a bit during the freeze-drying process. With more bioactive glass, the scaffold can maintain its porous structure better.

  1. I think some FTIR spectra should be listed. 

Authors Reply: Thank the reviewer for the suggestion. The FTIR spectra of regenerated cellulose, pure bioactive glass and the BG/cellulose scaffold are compared in Figure 3.

  1. Figure 2. There are two b.1. Please replace one with b.2.

Authors Reply: Thank the reviewer for the comment. The error of two b.1 has been corrected. Figure 2 is Figure 4 in the revised manuscript.

  1. Figure 4. P-value should be added, since these data must be conducted on the same day.

Authors Reply: P-value is added in Figure 8 in the revised manuscript.

Reviewer 2 Report

Overall the paper is well written but some corrections have to be made in order to make it clearer and reproducible (please see comments below) . The introduction could be improved adding some more references. In particular, it is necessary to add a deep discussion that compares the presented results with similar scaffolds (cellulose and biomaterials), to highlight the advantages or drawbacks of the materials employed. The bibliography is very poor, therefore could be improved with the discussion suggested. Some images are missing from Supplementary material.

Lines 107 and 108: pattern(c) (2CELL-1BG) and  pattern(d) (1CELL-1BG) demonstrated.

Comment:  there is an error, the (2CELL-1BG) is pattern (d) and (1CELL-1BG) is pattern (e); moreover

Fig 1: the symbol of the spade beside BG in pattern (a) is not reported in the legend above          

Line 113: 3.1.2. Mechanical Tests

Comment: change as 3.2

Fig 2: (b.1) is repeated twice and (b.2) is not written

Moreover  it has never been reported how many specimens were tested (n=?) for the compressive or tensile tests, since an average and standard deviation are reported in the text, please add n=… in the figure legend and in materials and methods.

Line 137 3d printed

Comment:3D

Line 139: All scaffolds have ordered interconnected porous structures with approximately 200 μm fiber spacing

Comment: Fig 3 a1 seems to show smaller pore size with respect to b1,c1,d1…please comment on these differences and mention that this aspect is reported in the supplementary material (Fig S2)

Line 145: Figure S1(a3)…and The SEM images in Figure S1(b3, c3 and d3)

Comment: there are no such figures in Supplement material but only  S1 (a, b, c, d)

Line 150: surfaces after 3-day simulated body fluid (SBF) immersion.

Comment: please add that the Protocol of hydroxyapatite formation on scaffolds in simulated fluids is reported in the supplementary material.

Line 155: EDS

Comment: please specify this acronym

Line 170: lose:BG=1:1 (c3, d4)……..

Comment: c3 is an error, change with d3

Line 182: The is because……

Comment:   something is missing in this sentence

Line 182: Specifically,, the………

Comment: delete one comma

Line 183: P<0.05, marked with * in Figure 4)…….

Comment: a statistical analysis has never been mentioned in materials and methods, please add what kind of test has been performed

Line 185: regrading…….

Comment:  regarding

Line 188: figure legend is in bold….

Line 189 : 3.2. Mechanical Tests……

Comment:  it must be deleted from figure legend

Line 195: human trabecular bone (2-12 MPa).

Comment:  please add the refernce where the range has been taken from

Supplement material:

line 1 pag 4: 84.3 ± 2.3% (2BG-CELL), and 85.2 ± 1.2% (SI),…

Comment: what is (SI)?

Author Response

Overall the paper is well written but some corrections have to be made in order to make it clearer and reproducible (please see comments below) . The introduction could be improved adding some more references. In particular, it is necessary to add a deep discussion that compares the presented results with similar scaffolds (cellulose and biomaterials), to highlight the advantages or drawbacks of the materials employed. The bibliography is very poor, therefore could be improved with the discussion suggested. Some images are missing from Supplementary material.

 Authors Reply: Thank the reviewer for the suggestions. The manuscript is revised according to the suggestions.

Lines 107 and 108: pattern(c) (2CELL-1BG) and  pattern(d) (1CELL-1BG) demonstrated.

Comment:  there is an error, the (2CELL-1BG) is pattern (d) and (1CELL-1BG) is pattern (e); moreover

Fig 1: the symbol of the spade beside BG in pattern (a) is not reported in the legend above           

 Authors Reply: Thank the reviewer for the comment. The editor uploaded the wrong manuscript. The errors in the revised manuscript were corrected and sent to the editor. The XRD patterns are presented in Figure 2 in the latest revised manuscript. The XRD pattern of BG is added.

Line 113: 3.1.2. Mechanical Tests

Comment: change as 3.2

Authors Reply: “3.1.2 Mechanical Tests” is changed to “3.4. Mechanical Tests” on line 288 page 7 in the revised manuscript.

Fig 2: (b.1) is repeated twice and (b.2) is not written

Authors Reply: The error in Fig 2 regarding (b.1) is corrected, which is Figure 4 in the revised manuscript.

Moreover  it has never been reported how many specimens were tested (n=?) for the compressive or tensile tests, since an average and standard deviation are reported in the text, please add n=… in the figure legend and in materials and methods.

 Authors Reply: Authors described that each type of scaffolds was tested five times using the selected typical samples, which means n=5 on line 200 page 4 in the revised manuscript. Authors add “n=5” in the sentence and in the figure legend now.

Line 137 3d printed

Comment:3D

 Authors Reply: The error is corrected.

Line 139: All scaffolds have ordered interconnected porous structures with approximately 200 μm fiber spacing

Comment: Fig 3 a1 seems to show smaller pore size with respect to b1,c1,d1…please comment on these differences and mention that this aspect is reported in the supplementary material (Fig S2)

 Authors Reply: Figure 3a1 show the pure cellulose samples, which underwent a bit deformation after freeze drying. Cellulose samples also swell in water, which is obvious different from the samples with the BG content. Such information is added in the revised manuscript on line 338-341, page 9.

Line 145: Figure S1(a3)…and The SEM images in Figure S1(b3, c3 and d3)

Comment: there are no such figures in Supplement material but only  S1 (a, b, c, d)

 Authors Reply: All Figures in Supplement materials are transferred to the revised manuscript. The orders of the Figure are rearranged.

Line 150: surfaces after 3-day simulated body fluid (SBF) immersion.

Comment: please add that the Protocol of hydroxyapatite formation on scaffolds in simulated fluids is reported in the supplementary material.

 Authors Reply: The Protocol of hydroxyapatite formation on scaffolds in simulated fluid is added in section 2.4.4, on line202-212 page 5.

Line 155: EDS

Comment: please specify this acronym

 Authors Reply: This acronym of EDS is given on line 175 page 4. It is energy-dispersive X-ray spectroscopy (EDS).

Line 170: lose:BG=1:1 (c3, d4)……..

Comment: c3 is an error, change with d3

 Authors Reply: Thank reviewer for the comment. The error is corrected in the revised manuscript on line 384 page 10.

Line 182: The is because……

Comment:   something is missing in this sentence

 Authors Reply: The reviewer is right. A typo happened here. The sentence is changed to “This is because…” in the revised manuscript.

Line 182: Specifically,, the………

Comment: delete one comma

 Authors Reply: One comma is deleted in the revised manuscript.

Line 183: P<0.05, marked with * in Figure 4)…….

Comment: a statistical analysis has never been mentioned in materials and methods, please add what kind of test has been performed

 Authors Reply: Thank the reviewer for the suggestion. The description of the statistical analysis is given in materials and methods section. The sentence is added: “All statistical data are expressed as the means with standard deviations. LSD analysis was used to compare pairs after one-way analysis of variance (ANOVA). Statistical significance was defined as *p < 0.05, **p < 0.01, and ***p < 0.001.”

Line 185: regrading…….

Comment:  regarding

 Authors Reply: The error is correted in the revised manuscript on line 415 page 11.

Line 188: figure legend is in bold….

 Authors Reply: The bold legend is corrected.

Line 189 : 3.2. Mechanical Tests……

Comment:  it must be deleted from figure legend

 Authors Reply: Thank reviewer for the comment. The error is corrected in the revised manuscript.

Line 195: human trabecular bone (2-12 MPa).

Comment:  please add the refernce where the range has been taken from

 Authors Reply: The reference is added in the Introduction part, line 35 page 1.

Supplement material:

line 1 pag 4: 84.3 ± 2.3% (2BG-CELL), and 85.2 ± 1.2% (SI),…

Comment: what is (SI)?

Authors Reply: ”SI” is a typo. This error is corrected. “SI” is changed to “1CELL-1BG” on line 386 page 10.

Round 2

Reviewer 1 Report

All of my comments were clearly addressed. Thus, I think it can be accepted in current form.

It will be better if checking the English writing.

Author Response

All of my comments were clearly addressed. Thus, I think it can be accepted in current form.

Authors Reply: Thank the reviewer for the suggestion.

Reviewer 2 Report

Line 58 polyhyhroxyalkanoates....?

Line 234: size range of 100 nm – 10 μm…….

Comment: 10 is still wrong…? should be 1 as in the line below?

Fig 1c: is the scale bar really 30 μm? Some particles look very big compared with the size reported in the graph 1d

only a few english language imperfections here and there, better to be revised by english mothertongue

Author Response

Line 58 polyhyhroxyalkanoates....?

 Authors Reply: The error has been corrected to “polyhydroxyalkanoates”.

Line 234: size range of 100 nm – 10 μm…….

Comment: 10 is still wrong…? should be 1 as in the line below?

Fig 1c: is the scale bar really 30 μm? Some particles look very big compared with the size reported in the graph 1d

Authors Reply: Thank the reviewer for the comment. The size range is actually between 100 nm and 10 μm. Authors add the histogram of the whole size range of the BG particles in Figure 1d. The main size is less than 1 μm. In Figure 1c, the scale bar is really 30 μm. Figure 1e is the histogram of the particle size distribution between 100 nm and 1 μm.